# The Role of SCAP/SREBP as Central Regulators of Lipid Metabolism in Hepatic Steatosis

**DOI:** 10.3390/ijms25021109

**Published:** 2024-01-16

**Authors:** Preethi Chandrasekaran, Ralf Weiskirchen

**Affiliations:** 1UT Southwestern Medical Center Dallas, 5323 Harry Hines Blvd., Dallas, TX 75390, USA; Preethi.Chandrasekaran@utsouthwestern.edu; 2Institute of Molecular Pathobiochemistry, Experimental Gene Therapy and Clinical Chemistry (IFMPEGKC), Rheinisch-Westfälische Technische Hochschule (RWTH) University Hospital Aachen, D-52074 Aachen, Germany

**Keywords:** SCAP, SREBP1c, de novo lipogenesis, liver, MASLD, pathogenesis, lipid metabolism, insulin resistance, fatty acid synthesis, triglycerides, therapy

## Abstract

The prevalence of metabolic dysfunction-associated steatotic liver disease (MASLD) is rapidly increasing worldwide at an alarming pace, due to an increase in obesity, sedentary and unhealthy lifestyles, and unbalanced dietary habits. MASLD is a unique, multi-factorial condition with several phases of progression including steatosis, steatohepatitis, fibrosis, cirrhosis, and hepatocellular carcinoma. Sterol element binding protein 1c (SREBP1c) is the main transcription factor involved in regulating hepatic de novo lipogenesis. This transcription factor is synthesized as an inactive precursor, and its proteolytic maturation is initiated in the membrane of the endoplasmic reticulum upon stimulation by insulin. SREBP cleavage activating protein (SCAP) is required as a chaperon protein to escort SREBP from the endoplasmic reticulum and to facilitate the proteolytic release of the N-terminal domain of SREBP into the Golgi. SCAP inhibition prevents activation of SREBP and inhibits the expression of genes involved in triglyceride and fatty acid synthesis, resulting in the inhibition of de novo lipogenesis. In line, previous studies have shown that SCAP inhibition can resolve hepatic steatosis in animal models and intensive research is going on to understand the effects of SCAP in the pathogenesis of human disease. This review focuses on the versatile roles of SCAP/SREBP regulation in de novo lipogenesis and the structure and molecular features of SCAP/SREBP in the progression of hepatic steatosis. In addition, recent studies that attempt to target the SCAP/SREBP axis as a therapeutic option to interfere with MASLD are discussed.

## 1. Introduction

MASLD is an umbrella term for a broad continuum of manifestations characterized by histological findings of macrovescicular hepatic steatosis of more than 5% of the hepatocytes in individuals who consume little or no alcohol [1]. This benign liver disease may progress to cirrhosis in more than 25% of the patients [2]. MASLD is further divided into metabolic dysfunction-associated fatty liver (MAFL), which is characterized by steatosis with or without mild lobular inflammation, and metabolic dysfunction-associated steatohepatitis (MASH), which is characterized by diffuse lobular inflammation and fibrosis and has great potential to progress into hepatocellular carcinoma [3]. MASLD and dyslipidemia are driven by common molecular mediators and, further, MASLD is strongly associated with dyslipidemia, representing one of the most common manifestations of the metabolic syndrome. The most important factor playing a significant role in the pathogenesis of MASLD is the crosslink between liver lipid metabolism and peripheral fat metabolism [4].

The three sterol element binding proteins (SREBPs), SREBP1a, SREBP1c, and SREBP2, are involved in a myriad of physiological and pathophysiological processes ranging from canonical functions, such as transcriptional regulation of fatty acid biosynthesis, endoplasmic reticulum (ER) stress formation, apoptosis, and autophagy, highlighting their distinctive role in lipid synthesis and metabolism [5]. SREBP cleavage activating protein (SCAP), an ER sensing protein, plays an inevitable role in the regulation of SREBP activity by transporting SREBP from the ER to the Golgi apparatus, facilitating the release of the *N*-terminus of SREBP by site-1 and site-2 proteases and permitting its entry into the nucleus. In this compartment, the *N*-terminal domain activates the transcription of several genes involved in lipid synthesis, such as HMG-CoA reductase, HMG-CoA synthase, fatty acid synthase, and glycerol-3-phosphateacyltransferase, when cellular cholesterol levels are low [6]. On the other hand, when cholesterol is present in large quantities in the ER membrane and exceeds the sharp threshold of 4–5% of total lipids, it binds to SCAP, thereby triggering a conformational change promoting SCAP interaction and formation of a ternary complex with the ER membrane retention protein insulin-induced gene 1 (INSIG1), which is composed of 277 amino acids, or insulin-induced gene 2 (INSIG2), which is composed of 225 amino acids. This reduces the transport of the SCAP/SREBP complex to the Golgi, thereby inhibiting the synthesis of cholesterol and initiating a negative feedback mechanism that maintains cholesterol homeostasis [7].

Consequently, SCAP plays a highly influential role in regulating cholesterol and triglyceride levels in the body, and, hence, in maintaining intracellular fatty acid homeostasis [6]. Multiple studies have illuminated the strong associations of MASLD with insulin resistance, obesity, and dyslipidemia, with de novo lipogenesis being a major factor involved in the development of hepatic steatosis in insulin resistance states [8].

To sum up, the role of SREBP1s and SCAP is pivotal in the pathogenesis of MASLD as the former is the main transcription factor involved in lipogenesis activation and the latter is the key mediator in the activation of SREBPs. It is thus crucial to gain a deeper understanding of SCAP/SREBP regulation in MASLD, which may open avenues for novel therapeutic drug designs targeting the SCAP/SREBP axis for treating fatty liver and hypertriglyceridemia [8]. To date, several drugs have been developed to inhibit lipid synthesis by targeting SCAP/SREBP activities in various metabolic disorders. Nevertheless, there is currently no effective treatment available for MASLD. The prevalence of MASLD in the obese/overweight population is as high as 79%, emphasizing the urgent need to unfold promising therapeutics targeting critical mediators, such as SCAP/SREBP, involved in the pathogenesis of MASLD [9]. This review provides comprehensive information on the association of SCAP/SREBP and de novo lipogenesis, its relation to hepatic steatosis, molecular features of SCAP/SREBP activity, and current studies and developments in targeting the SCAP/SREBP in the pharmacotherapy of hepatic diseases.

## 2. Structure of SCAP

SCAP functions as an ER-cholesterol-sensing membrane protein. It is composed of 1276 amino acids, with a molecular mass of 140 kDa, and is divided into two functional regions [10]. The first functional region includes the *N*-terminal transmembrane (TM) region, consisting of 735 amino acids and eight defined TM helices separated by hydrophilic loops. Loop 1 has cholesterol-sensing properties, seven of which are oriented towards the ER lumen while all other TM helices are faced toward the cytosol. Loop 6 contains a hexapeptide sequence (MELADL) that has an affinity for the coat protein complex II (COPII), which is relevant in the clustering of ER proteins into coated vesicles for transport to the Golgi apparatus [11]. The second functional region is the soluble C-terminal domain, consisting of 540 amino acids, extending into the cytosol, and including four distinct WD40 protein-interacting repeats, mediating the binding to SREBPs and enabling the formation of a stable SCAP/SREBP complex [12].

In short, SCAP can be classified as a protein with five functional subdomains: (i) loop 1, with the cholesterol-binding domain and the TM helices 2–6 that form the sterol sensing domain and contain the (ii) INSIG-binding domain, sharing homology with other cholesterol regulatory proteins (iii) the COPII-binding hexapeptide, recognized by a wide array of cargo proteins, such as SEC24, (iv) loop 7, that has affinity for loop 1, and (v) the carboxy-terminal domain (CTD) containing the four WD40 repeats mediating SREBP binding (Figure 1) [7].

Previous studies have shown that cholesterol binding to loop 1 triggers the dissociation of loop 1 from loop 7, inducing a conformational change that increases the affinity of SCAP for INSIGs and the COPII hexapeptide (MELADL) sequence motif [13]. In other words, cholesterol binding to loop 1 causes conformational changes, disrupting the interaction between loops 1 and 7, leading to sequestration of the MELADL motif, which marks the C-terminus of the sterol sensing domain and prevents it from being recognized by COPII proteins [14].

To elaborate further, SCAP has eight TM helices and INSIG has six TM helices, respectively. The TM helices of SCAP on the cytosolic side are connected by short loops, except for loop 6 which is located between TM 6 and TM 7, and has approximately 94 residues [15]. All luminal loops in SCAP, except loop 1 located between TM 1 and TM 2 (~239 residues) and loop 7 between TM 7 and TM 8 (~176 residues), are short. On the other hand, all TM helices in INSIG are connected by short loops, suggesting that these TM helices dominate the contact between SCAP and INSIG [15].

The formation of dimers between SCAP and INSIG is the key control element in the regulation of cholesterol balance in mammalian cells [15]. When cholesterol levels in the cell exceed a certain threshold the SREBP2/SCAP complex is anchored to the ER through a sterol-dependent interaction between the sterol-sensing domain of SCAP and INSIG. This interaction is more sensitive to hydroxylated cholesterol derivatives, such as 25-hydroxycholesterol (25HC). Consequently, the structure of the SCAP/INSIG/25HC ternary complex is of major importance for the development of novel therapeutics. This ternary complex interacts with Sec23/24, which are basic functional units of COPII coat proteins, in a secretion-associated RAS-related GTPase 1 (SAR1)-dependent manner through an ER export signal located in the third cytoplasmic loop. The binding of the Sec23/24/SRA1 complex is prohibited when INSIG binds SCAP, subsequently preventing SCAP/SREBP binding [16]. The 25HC molecule is sandwiched between SCAP and INSIG2 in the luminal leaflet of the membrane and the unwinding of a central segment of SCAP in the middle of the membrane is a crucial step required for the formation of the 25HC binding pocket [17]. The preference of 25HC over cholesterol in establishing SCAP/INSIG interaction is explained by the fact that the binding site is mainly constituted by hydrophobic residues in TM3 and TM4 of INSIG2 and segment 4 to segment 6 in SCAP [17]. The 25OH group at the end of the 25HC tail is exposed to the cytosolic milieu through a hydrophobic cavity enclosed by SCAP and INSIG2. The limited contact between segment 5 of SCAP and INSIG TM4 causes a fenestration on one side, resulting in a cavity that exposes the 25-OH group of 25HC to the cytosol. The end of the iso-octanol tail creates a hydrophilic environment, explaining the preference of 25HC over cholesterol in facilitating SCAP/INSIG2 interaction. Thus, 25HC acts like a molecular glue stabilizing the unwound conformation of SCAP-S4, which is critical for INSIG association. This establishes a strong molecular basis for sterol sensing. The unique feature of this SCAP segment is that it breaks into two subhelices, one of which leans towards the interior of the sterol sensing domain creating space for the ligand [17].

The structure of SCAP/INSIG2 that is formed in the presence of sterols, such as 25HC, solved by cryogenic electron microscopy is depicted in Figure 2, illustrating the interaction of the TM regions and the locations of the sterol-sensing domain orientated to the ER lumen and the location of the COPII binding domain orientated to the cytosol.

## 3. Structure of SREBPs

Out of the three SREBPs that regulate the synthesis of fatty acids and cholesterol, SREBP1c and SREBP1a are translated from a single gene on human chromosome 17p11.2 through the usage of alternate transcription start sites producing alternate forms of exon 1, whereas SREBP2 is derived from a separate gene located on human chromosome 22q13 [18,19,20,21]. In other words, SREBP1c differs from SREBP1a in its *N*-terminus encoded by exon 1. It is important to note that SREBP1 isoforms in mice and humans are abundantly expressed in the liver, adrenal glands, skeletal muscle, white adipose tissue, and brain, whereas SREBP2 is expressed ubiquitously [22].

The SREBPs are synthesized as long precursors of 1123–1147 amino acids in length. The *N*-terminal part (~480 amino acids) contains a basic helix-loop-helix-leucine zipper. The middle segment, composed of 80 amino acids, has two membrane-spanning domains separated by a short hydrophilic sequence of 30 amino acids. The C-terminal segment consists of 590 amino acids (Figure 3A) [18,23]. Unlike other proteins of the basic helix-loop-helix-leucine zipper family that recognize symmetric E-boxes, SREBPs have a tyrosine instead of a conserved arginine in their basic regions, which allows the recognition of asymmetric sterol responsive elements (SRE) [24]. These non-palindromic SREs are regulatory elements common in the HMG-CoA synthase and LDL receptor gene promoters.

As the first step to initiate the transcription of genes involved in cholesterol and fatty acid biosynthesis a protease recognition site in SREBP (site 1), which is in the middle of the luminal loop in SREBP, is cleaved by a protease, leading to the breakdown of covalent bonds between two membrane-spanning domains, followed by a second protease cleavage at site 2, which is located in the middle of the first membrane-spanning segment. Consequently, this leads to the release of the NH_2_ terminal segment, permitting its entry into the nucleus and allowing binding of the SREs of promoters encoding enzymes involved in cholesterol biosynthesis and LDL receptors (Figure 3B) [25].

**Figure 3 ijms-25-01109-f003:**
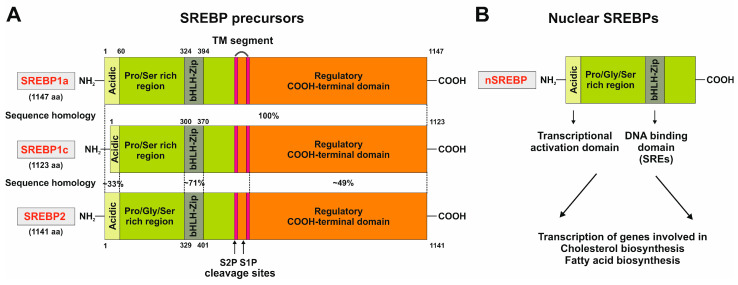
Domain structure of the three SREBP family members SREBP1a, SREBP1c, and SREBP2. (**A**) The precursors of the three members of the SREBP family are comprised of three domains, namely, an NH_2_-terminal domain composed of an acidic domain, a proline and serine-rich stretch, and a basic helix-loop-helix-leucine zipper domain, two hydrophobic transmembrane segments separated by a short loop of 30 amino acids, and a carboxy-terminal regulatory region. This part of the figure was redrawn in modified and extended form from [26]. (**B**) The processed nuclear SREBP (nSREBP) contains only the *N*-terminal part of the precursor, composed of the acidic domain acting as a transcriptional activation domain and the basic helix-loop-helix-leucine zipper motif that binds sterol response elements (SREs) and acts as a DNA-binding domain. For more details see text.

In human SREBP1a, exon 1, encoding the NH_2_ terminus, is composed of 29 amino acids including eight negatively charged amino acids [18]. This sequence is part of an acidic, 42 amino acid spanning region, in which a domain of 12 amino acids is required for transcriptional activation. This amino acid stretch includes four negatively charged amino acids, encoded by exon 2. On the other hand, exon 1 of SREBP1c has only five amino acids, out of which only one is negatively charged. This is part of a 24 amino acid activation domain containing six negatively charged amino acids encoded by exon 2 [18]. This raises the possibility of SREBP1c being a less potent transcription factor compared to SREBP1a, which was tested and proven in one of the landmark studies by Shimano and colleagues [27].

A cluster of putative binding sites for several transcription factors harboring a nuclear factor Y (NF-Y) site, an E-box sequence, an SRE and an SP1 binding site is at −90 bp of the SREBP1c promoter [28,29]. In particular, it was suggested that the NF-Y site and the SRE are majorly responsible for SREBP activation [28]. The sequence motifs in the murine SREBP1 promoter include SRE (5′-TCACNCCAC-3′) and E-box (5′–CANNTG-3′) [28,30]. The domain structure of the three members of the SREBP family, and the steps involved in the proteolytic activation of SREBPs, are illustrated in Figure 4.

## 4. Regulation of De Novo Lipogenesis by SCAP/SREBP1c

Under normal physiological conditions, the SREBPs are central and ubiquitous modulators of lipid homeostasis [31]. The controlled transport of SREBP from the ER to the Golgi is an essential step in the regulation of cholesterol synthesis and other membrane lipids in mammalian cells. Therefore, SCAP has a crucial dual role as a sterol-sensing protein and as a cleavage-activating protein that is required for the transport of SREBPs from the ER to the Golgi apparatus. In sterol-depleted cells, the SCAP-mediated transport of SREBP to the Golgi apparatus is followed by the release of an amino-terminal domain of SREBP that enters into the nucleus and activates lipogenic target genes [32]. In contrast, in sterol-overloaded cells, cleavage at site 1 is abolished which in turn halts the cleavage at site 2 and, as a result, SREBP remains bound to membranes and is prevented from being transported to the Golgi, thereby declining the transcription of the target genes. This feedback mechanism is extremely important in maintaining cholesterol homeostasis [33].

Moreover, the transcriptional regulation of SREBP1c and SREBP2 involves a feed-forward loop in which a feed-forward regulation is mediated by sterol-responsive elements present in the promoters of each gene, promoting their own expression [29,33].

The selective regulation of SREBP1c, which is the abundant isoform in liver and adipose tissue, occurs by three factors, namely, liver X receptor-α (LXRα), LXRβ, and nuclear receptors, forming heterodimers with retinoid X receptors that are activated by various sterols [34,35,36]. The SREBP1c promoter has an LXR binding site that activates the transcription of SREBP1c in the presence of LXR agonists, indicating that LXR plays a key role in the insulin-dependent stimulation of SREBP1c [35]. The enzymes catalyzing the hepatic fatty acid synthesis, acetyl-CoA-carboxylase (ACC) and fatty acid synthase (FAS), are controlled at the transcriptional level via SREBP1c activation by insulin and via carbohydrate-responsive element-binding protein (ChREBP) that is activated by glucose metabolites. This network in turn is controlled by LXR, which directly induces the expression of ACC and FAS. Noteworthy, the co-existence of hyperinsulinemia markedly increases de novo lipogenesis in MASLD by activation of the insulin receptor, subsequently activating several signaling cascades that lead to the promotion of lipogenic genes including ACC and FAS, and thereby triggering the synthesis of sterols and fatty acids [37,38].

From the three SREBP isoforms, SREBP1a is the most potent activator of all SREBP-responsive genes involved in cholesterol, fatty acid, and triglyceride synthesis, while SREBP1c has a more selective role in enhancing fatty acid synthesis and not cholesterol synthesis. SREBP2 preferentially activates genes involved in cholesterol synthesis, such as HMG-CoA reductase, HMG-CoA synthase, and squalene synthase, that catalyze the biosynthesis of the cholesterol precursor squalene [39].

Numerous studies have reported multiple regulatory signals, such as insulin and growth factors, unfolded protein response, and altered intracellular concentration of K^+^ modulating the transcriptional and post-transcriptional maturation and stability of SREBPs [25]. For instance, one study showed that the mammalian target of rapamycin complex 1 (mTORC1) is sufficient to achieve stimulation of specific metabolic pathways by activation of SREBP1 and SREBP2 [40].

Initially, the insulin signaling pathway starts with the receptor-mediated tyrosine phosphorylation of insulin receptor substrate 1 and 2 (IRS1/IRS2), followed by IRS activation of PI3K, which in turn converts phosphatidylinositol-4,5-bisphosphate (PIP2) to phosphatidylinositol-3,4,5-trisphosphate (PIP3). The PIP3 interacts with protein kinase B (AKT/PKB) and recruits it to the membrane where it is activated. It is well known that insulin regulates SREBPs at various levels including stability, proteolytic processing by decreasing the INSIG2 expression, promoting mTORC1-induced p70 S6 kinase function, and the transcription of SREBP mRNA [41]. In line, SREBP1c levels are reduced in mice during fasting states with low insulin content, which supports the fact that insulin strongly induces SREBP1c transcription [42].

As discussed above, SREBP1 and SREBP2 are regulated by mTORC1. Dephosphorylated lipin1, which is a substrate of mTORC1, inhibits nuclear localization of SREBPs in NIH-3T3 cells, while Lipin 1 carrying a mutation at the mTORC1 phosphorylation site suppresses nuclear entry for SREBP1c and expression of lipogenic target genes [43].

Another interesting mechanism was found in mice, in which the CREB-regulated transcription co-activator 2 (CRTC2) was found to function as a mediator of mTORC1 signaling and a modulator of COPII-dependent SREBP1 processing [44]. Together, insulin activates SREBP1 via the AKT/PKB pathway by inhibiting glycogen synthase kinase-3 (GSK3), enhancing the stability of SREBP1, and increasing the expression of SREBP1 through mTORC1 activation [45].

Moreover, there is growing evidence that the concentration of several substances, such as sphingomyelin or polyunsaturated fatty acids (PUFAs), affects the protein processing of SREBPs. For example, it has been shown that the depletion of sphingomyelin in CHO cells blocks the proteolysis of SREBPs [46]. Conversely, PUFAs inhibit the proteolytic processing of SREBPs in murine livers [47], ultimately reducing cholesterol synthesis and uptake (Figure 5).

The nuclear liver X receptors (LXRs) play a crucial role in regulating fatty acid metabolism. They activate the expression of SREBP1c and PPARα, which promote fatty acid β-oxidation. Early studies have shown that in fasted states PPARα is primarily responsible for fatty acid oxidation to acetyl-CoA and ketone bodies, while the expression of SREBP1c is reduced. Conversely, in refed states, increased SREBP1c leads to lipogenesis, while PPARα is reduced. This evidence suggests that there is cross-talk between these two transcriptional factors and their coordinated reciprocal regulation is critical for the nutritional regulation of fatty acids and triglycerides. Recent studies have also indicated that PPARs suppress SREBP1c promoter activity through LXR response elements. This is supported by the repression of LXR/RXR activation of LXRE containing the SREBP1c promoter in hepatocytes overexpressing PPARs [49].

## 5. The Central Role of SCAP/SREBP1c in MASLD Pathogenesis

The widely accepted ‘multi-hit’ theory of MASLD pathogenesis was derived from early studies suggesting excessive fatty acids contributing to hepatic steatosis and insulin resistance as the first hit and subsequent damage of hepatocytes due to inflammation, fibrosis, and other pathological changes due to oxidative stress, lipid peroxidation, ER stress, and lipotoxicity as the second hit [50]. In this scenario, it is obvious that the imbalance between lipid synthesis, availability, and lipid disposal by fatty acid oxidation and VLDL secretion is the fundamental cause of steatosis [51].

The three main sources of fatty acids in the liver include hepatic de novo lipogenesis, free fatty acids from adipose tissue, and the uptake of dietary fats (Figure 6) [52]. The hallmark feature of insulin resistance and obesity states in MASLD is that the content of hepatic fatty acids derived from de novo lipogenesis contributes about 26%, in contrast to 5% in normal healthy individuals [52].

Increased rates of lipogenesis were shown to be significant contributors to liver lipid accumulation in multiple insulin-resistance rodent models [53]. The excessive accumulation of triglycerides in the liver is mainly due to the increased availability of free fatty acids through lipolysis of insulin resistant and hypertrophied adipose tissue. In addition, increased fatty acid synthesis leads to decreased mitochondrial fatty acid oxidation due to the inhibition of carnitine palmitoyl transferase 1 by malonyl-CoA and the increased release of free fatty acids through TG lipolysis due to increased lipase activity [8]. The development and progression of MASLD is majorly driven by alterations in lipid profiles induced by SREBP activation or repression and alteration of different signaling pathways, such as PI3K-PKB-AKT-mTORC1 [54].

SREBP1c is a critical modulator involved in the regulation of hepatic de novo lipogenesis by insulin by preferentially activating genes involved in fatty acid synthesis, such as fatty acid synthase, elongation of very long chain fatty acids-like 6 (ELOVL6), ATP-citrate lyase, acetyl CoA carboxylase, malic enzyme, glucose 6 phosphate dehydrogenase, stearoyl-CoA-desaturase, and glycerol-3-phosphate acyl transferase, whereas SREBP2 preferentially activates genes required for cholesterol synthesis [55]. SREBP1c binds to the promoter of the patatin-like phospholipase domain containing protein 3 (*PNPLA3*) and regulates its expression. This enzyme acts as a triacylglycerol lipase that mediates the hydrolysis of triacylglycerol in adipocytes, and thus, hepatic lipid accumulation is stimulated by the accumulation of PNPLA3 in the lipid droplets of hepatocytes, promoting hepatic fat accumulation [56].

The cleavage and transcriptional activity of SREBP are inhibited by AMP-activated protein kinase (AMPK). This conserved enzyme is an energy sensor playing a key role in cellular energy homeostasis and its stimulation suppresses SREBP cleavage and nuclear translocation attenuating hepatic steatosis, as demonstrated in LDL-deficient mice with diet-induced insulin resistance [57].

Recently, the impact of microRNAs on SREBP expression affecting intracellular lipid levels, HDL formation, and cholesterol transport, as well as potential associations between long non-coding RNAs (lncRNAs) and MASH by regulating hepatic lipogenesis, have become the focus of research [58]. In particular, it was demonstrated that lncARSR promotes hepatic lipogenesis via the AKT-SREBP1c pathway [59].

Similarly, ER stress promotes lipogenesis by activating SREBP1c, leading to the progression of MASLD and severe forms of MASH. The unfolded protein response (UPR) is linked to inflammation and hepatocyte apoptosis in MASH, and this claim is supported by the finding that hepatic overexpression of the heat-shock protein-70 family member glucose regulated protein 78 (GRP78), which is critically implicated in the folding and assembly of proteins in the ER, reduced markers of ER stress in ob/ob mice by inhibiting the cleavage of SREBP1c and SREBP2 target genes [60]. In a similar context, adipogenesis is upregulated through the IRE1-XBP1 and PERK-PEIF2α axis, while, on the other hand, interaction between ATF6, SREBP2, and HDAC1 limits adipogenesis [60]. Additionally, ceramide synthase and its derivatives modulate ER stress and MASLD progression by decreasing INSIG1 and regulating SREBP1c cleavage [61].

Another factor relating SREBP to the pathogenesis of MASLD is lipotoxicity in hepatocytes, which provokes hepatocyte damage when lipotoxic substances are elevated beyond the hepatocyte’s ability to transport them. For example, when bile acids disrupt the cell membranes, free cholesterol activates the SREBP2 upregulation of LDL receptors, thereby reducing the biotransformation of cholesterol to bile acids. These lipotoxic substances accumulate and cause inflammation, fibrosis, and progression to MASH [62]. Among the dietary factors, fructose plays a significant role in the development of MASH by inducing hepatic de novo lipogenesis by activating key transcription factors, such as SREBP and ChREBP [63].

Liver fibrosis is characterized by the excessive deposition of extracellular matrix components between hepatocytes and sinusoids, causing liver stiffness and distortion of the liver architecture. Activated hepatic stellate cells in the liver produce collagen, which is an important component of the extracellular matrix network. TGF-β activates signaling pathways, such as MAPK, mTOR, PI3K/AKT, and Rho/GTPase, leading to liver fibrosis. These pathways are regulated by SREBP1c, which impacts hepatic stellate cell activation [64]. Furthermore, studies highlighting the mechanisms by which SREBP1c regulates hepatic stellate cells and liver fibrosis have revealed that the overexpression of SREBP1c inhibits liver fibrosis in mice by reducing TGF-β levels and signaling through SMAD3 and AKT1, AKT2, and AKT3 [65].

Strikingly, SREBP2 has also been implicated in liver fibrosis through the regulation of cholesterol levels in hepatic stellate cells. For example, in mouse models of hepatic steatosis and in primary mouse hepatic stellate cells, the nuclear form of SREBP2 increases with hepatic stellate cell activation [66]. Therefore, it is appropriate to define the SREBPs as pro-fibrotic mediators, as they activate the TGF-β lipotoxicity-induced development of liver fibrosis.

## 6. Current Studies on SCAP

Currently, there is no approved or well-defined treatment strategy to treat MASLD. Therefore, detailed information and scientific advances in preclinical and basic research are urgently needed to develop suitable pharmacotherapies to combat MASLD and secondary diseases resulting thereof.

Noteworthy, the overexpression of SREBP1c in a transgenic mouse model resulted in hepatic steatosis due to increased de novo lipogenesis [27]. It was demonstrated in the same study that SREBP1c is a comparatively weaker transcription factor than SREBPa. This was elucidated by comparing SREBP1a and SREBP1c transgenic mice, where there was a massive liver enlargement in SREBP1a transgenes due to increased accumulation of triglycerides and cholesterol, while the livers of SREBP1c transgenic mice were only slightly enlarged, with a moderate increase in triglycerides and no change in cholesterol. This was also confirmed by in vitro studies [27].

One other striking finding demonstrated that deletion of SREBP1c in the liver of ob/ob mice resulted in a 50% reduction of hepatic triglycerides, portraying the unique role of SREBP1c in hepatic steatosis exhibited by insulin resistance [67]. In addition, it was shown that the liver-specific deletion of SCAP resulted in a 90% reduction of fatty acid synthesis rates and prevented hepatic steatosis in ob/ob mice models [67]. In line, another study by Horton and colleagues demonstrated that the overexpression of SREBP1c in the adipose tissue of diabetic mice resulted in hepatic steatosis, hyperglycemia, and hyperinsulinemia. This correlated with a 4-fold increase in the fatty acid synthesis rate, and a 2–6-fold higher rate of mRNAs encoding lipogenic genes, such as FAS, glycerol-3-phosphate acyltransferase, acetyl-CoA carboxylase, and glucose-6-phosphate dehydrogenase [53]. However, the nuclear SREBP2 levels and mRNA levels for genes involved in cholesterol homeostasis remained unchanged in the livers of these mice, suggesting that the amount of nuclear SREBP1 (nSREBP1c) critically contributes to the fatty liver phenotype in these diabetic mice, most likely by the transcriptional activation of genes responsible for lipogenesis [53].

An important in vitro study demonstrated that mutant SCAP, generated by a single amino acid substitution (D443N) in the sterol-sensing domain, is resistant to inhibition by sterols. As a consequence, cells expressing this mutated variant overproduce cholesterol [68]. Similarly, transgenic mice expressing this mutant in the liver exhibited a similar phenotype, with elevated nSREBP1 and nSREBP2 stimulating the synthesis of cholesterol and fatty acids and thereby causing hepatic lipid accumulation. Therefore, this model strongly suggests that the mutant SCAP form causes resistance to sterol suppression, stimulates proteolytic procession of native SREBPs in the liver, and decreases the normal sterol-mediated feedback regulation of SREBP cleavage that is the key sensor for sterols [69].

Furthermore, the study by Shimano and colleagues supports the notion of distinct hepatic roles of SREBP1 and SREBP2, as evidenced by the death of most of the SREBP1 (both SREBP1a and SREBP1c) deficient mice in utero. Interestingly, the surviving mice exhibited a decrease in fatty acid synthesis and an increase in nSREBP2 as compensation for the loss of SREBP1 [70].

Multiple lines of evidence suggest that SREBP1c mediates the lipogenic action of insulin in the liver. One such study by Foretz and colleagues demonstrated an increase in SREBP1c mRNA and its target genes in insulin-treated rat hepatocytes [71]. The contribution of SREBP1c to glucose uptake and synthesis is well-established and is supported by the finding that the overexpression of SREBP1c in hepatocytes induces glucokinase (GCK) expression and suppresses the gluconeogenic enzyme phosphoenolpyruvate carboxykinase (PEPCK) [72].

Noteworthy, post-translational modifications also impact the overall activity of SREBPs. For example, neddylation of SREBP1c competes with its ubiquitination and promotes hepatic steatosis, while treatment with the neddylation-inhibitor Pevonedistat (MLN4924) decreased high fat diet-induced hepatic steatosis by decreasing SREBP1c and hepatic triglycerides [73].

Multiple studies support the role of lncRNAs in the development of hepatic steatosis, and one of them showed that lncRNA Gm15622 stimulates the expression of SREBP1c, promotes hepatic lipid accumulation, and is highly expressed in the livers of mice with high fat diet-induced obesity [74].

Recently, liver-specific gp78 knockout mice were generated by Liu and colleagues, and these mice showed an increase in INSIG1 and INSIG2, and a decrease in SREBP, which in turn decreased lipid biosynthesis. Thus, inhibition of gp78 may be a novel therapeutic avenue to treat MASLD and metabolic disorders [75].

One study illuminated the effects of phosphorylated Krüppel-like factor 10 (KLF10) in the regulation of MASLD through SREBP1c. Interestingly, SREBP1c was suppressed by phosphorylated KLF10 through promoter binding, suggesting altered KLF10 expression as an alternate approach to treating MASLD [76]. Another study by Akbari et al. aimed at evaluating the potential beneficial effects of *Capparis spinosa* extracts on MASH pathogenesis. *Capparis spinosa* represents a traditional plant previously used to treat dyslipidemia, reduce SREBP1c expression, and induce carnitine palmitoyltransferase 1 (CPT1) expression in MASH rat models, providing evidence for the favorable therapeutic effects of *Capparis spinosa* for ameliorating steatosis via modification of de novo lipogenesis and the β-oxidation pathway genes [77].

Finally, a decrease in ectopic fat deposition and meta-inflammation by reducing the hepatic stimulation of the interferon gene (STING)/NF-κB pathway activation was demonstrated in macrophage SCAP-specific null mice subjected to a high-fat and high-cholate diet (i.e., Paigen diet), which resulted in a lean MASLD phenotype. Thus, inhibition of macrophage SCAP can, in addition, be a promising treatment approach for lean MASLD [78].

## 7. Developments in SCAP as a Therapeutic Agent

Given the importance of SCAP/SREBP in the regulation of lipid metabolism, and its indispensable role in the pathogenesis of fatty liver, unraveling the multiple specific molecules and inhibitors targeting the SCAP/SREBP axis has become the focus of translational research (Figure 7).

Betulin is a small-molecule inhibitor with wide pharmacological potential. It has protective effects against cardiovascular diseases, cancer, diabetes, oxidative stress, and inflammation. In addition, it suppresses SCAP/SREBP translocation, thereby attenuating the biosynthesis of cholesterol and fatty acid [79]. Similarly, the phosphodiesterase inhibitor dipyridamole inhibited the nuclear accumulation of SREBP by selectively blocking SCAP/SREBP transport from the ER to the Golgi, retaining the INSIG-SCAP-SREBP complex in the ER [80]. Moreover, the binding of lycorine to SCAP suppressed the SREBP pathways without inducing ER stress. Instead, this small molecule promoted SCAP lysosomal degradation in a macroautophagy/autophagy-independent pathway that is completely distinct from current SCAP inhibitors [81].

Mechanistic studies of fatostatin, a compound that blocks the ER to Golgi movement of SCAP, thereby suppressing SREBP activation, showed that this cell-permeable diarylthiazole binds SCAP directly outside of the sterol-sensing domain and suppresses Golgi-specific glycosylation and ER to Golgi translocation. The efficacy of fatostatin has been proven in pre-clinical models of metabolic diseases [82].

A very recent study demonstrated that the global flavin-containing monooxygenase 2 (FMO2) impairs lipogenesis and protects against the progression of MASH by directly binding to SREBP1, thereby preventing its translocation and subsequent activation [83].

Similarly, HSP90 has been linked to lipid metabolism, and one study showed that HSP90β was overexpressed in MASLD patients and obese mice, while its depletion in mice promoted mature degradation of SREBPs and reduced cholesterol and neutral lipids. In line, the HSP90β selective inhibitor corylin suppressed AKT activity at Thr308 and specifically promoted the ubiqutination and proteasomal degradation of mature SREBPs and ameliorated fatty liver disease and atherosclerosis [84].

Lastly, natural product SREBP inhibitors have gained importance recently in treating various MASLD-associated risk factors, such as obesity, atherosclerosis, and diabetes. Arcitigenin, derived from *Arctium Lappa* L., increased AMPK acetyl-CoA carboxylase phosphorylation and upregulated the expression of downstream genes related to fatty acid β-oxidation, such as carnitine palmitoyltransferase 1 and acyl-CoA oxidase 1, but downregulated the expression of SREBP1c and lipogenesis-related genes [85]. Xanthohumol, derived from *Humulus lupulus*, blocks the expression of SREBP1c mRNA and the binding of SCAP/SREBP to COPII vesicles by binding to Sec23/24 [86,87]. Rohitukin, which can be isolated from the plant *Pimpinella anisum* L., commonly known as anise, was shown to decrease the expression of SREBP2 and its target genes [88]. Ligustrazine was shown to improve dyslipidemia by downregulation of the progestin and adipoQ receptor 3 (PAQR3) and inhibition of the SCAP/SREBP1c pathway in *ApoE*^−/−^ mice fed with a high fat diet [89,90].

Recently, multiple studies have been focusing on targeting SCAP in animal models of insulin resistance, with the potential to translate these findings into clinical trials aimed at reducing hepatic steatosis. Targeting SCAP could be more effective in alleviating hepatic steatosis because SCAP is a central regulator necessary for activating all three isoforms of SREBP. Evidence suggests that deleting SCAP eliminates the nuclear forms of both SREBP1 and SREBP2 in the liver. Additionally, when SCAP is deleted in the liver of a normal mouse, the expression of genes involved in fatty acid and cholesterol synthesis is reduced, leading to a 70% decrease in fatty acid synthesis and an 80% decrease in cholesterol synthesis rates.

In summary, deleting SCAP prevents SREBP activation and the expression of genes required for fatty acid synthesis, resulting in the inhibition of de novo lipogenesis. This makes SCAP a key therapeutic target for resolving hepatic steatosis.

## 8. Conclusions

De novo lipogenesis, the metabolic pathway that synthesizes saturated fatty acids and monounsaturated fatty acids from acetyl-CoA, only accounts for a small fraction of fatty acids in the liver in a healthy human population. In lean individuals, this fraction is typically around 5%. However, in individuals with MAFLD, the rate of de novo lipogenesis is significantly increased, with 25% of triglycerides arising from de novo lipogenesis. The expression of de novo lipogenesis enzymes in the liver is influenced by dietary lipids and nutritional state. However, the lipogenic transcription factors, and, consequently, DNL, are constantly active. The chronic energy excess caused by both glucose flux and hyperglycemia acts as a driving force in elevating triglyceride synthesis from saturated fatty acids and monounsaturated fatty acids. Therefore, the induction of de novo lipogenesis through SREBP1 suggests a strong link between de novo lipogenesis, SREBP1, and the progression of MAFLD [91].

MASLD progression to MASH has a multitude of detrimental effects and is one of the leading causes of liver disease worldwide affecting one-third of the world’s population. Multiple studies have documented the central role of the SCAP/SREBP axis in regulating lipid homeostasis. Thus, targeting the SCAP/SREBP axis may be an attractive and powerful option in treating MASLD. SCAP, a cholesterol sensor, impacts MASLD and other metabolic diseases. It is an indispensable protein required for the transport and activation of all three SREBP isoforms. SCAP and SREBPs are intricately linked to numerous signaling pathways, contributing immensely to the pathogenesis of MASLD. Therefore, pharmacological modulation of SCAP/SREBP activation by small-molecule inhibitors can be an attractive option for treating this complex disease with its wide spectrum of clinic-pathological conditions.

## Figures and Tables

**Figure 1 ijms-25-01109-f001:**
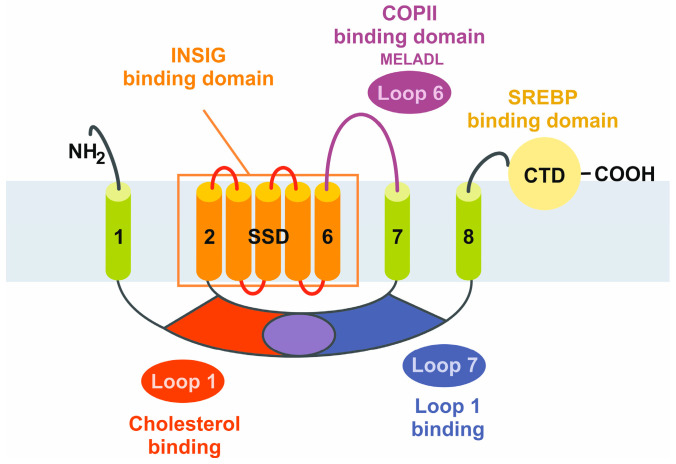
Schematic structure of the sterol element binding protein cleavage activating protein (SCAP). The SCAP protein is composed of five functional subdomains that are relevant for the binding of cholesterol, insulin-induced genes (INSIGs), coat protein complex II (COPII), and SREBPs, or for sensitizing cholesterol concentrations. The sterol-sensing domain (SSD) mediates the responsiveness towards cholesterol. The eight transmembrane domains are numbered from the N-terminus to the C-terminus. CTD, carboxy-terminal domain.

**Figure 2 ijms-25-01109-f002:**
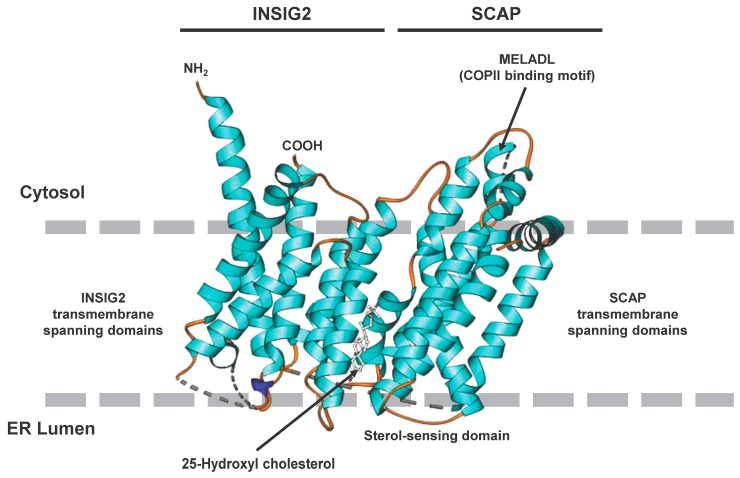
Cryogenic electron microscopic structure of the transmembrane region of the human SCAP and INSIG2 complex formed in the presence of 25-hydroxycholesterol. The sterol is sandwiched in a hydrophobic central pocket formed between SCAP and INSIG2 in the luminal leaflet of the membrane. The transmembrane spanning domains of INSIG2 and SCAP anchor the complex in the endoplasmic reticulum membrane, while the COPII binding site, which includes the MELADL motif, is exposed to the cytosol. More details about the complex resolved at resolutions of 3.3 to 3.9 Å are given elsewhere [17]. The image was generated with Ribbons XP Version 3.0 using coordinates deposited in the RCSB Protein Data Base (access. no.: 6M49).

**Figure 4 ijms-25-01109-f004:**
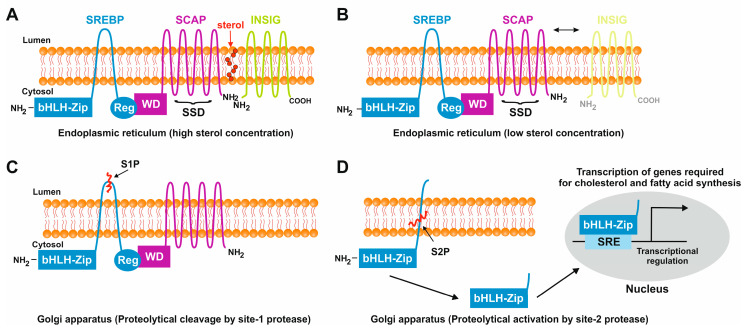
Sterol-mediated proteolytic activation of SREBPs. (**A**) In conditions with abundant sterol content, SREBP, SCAP, and INSIG are anchored in the membranes of the endoplasmic reticulum. Under these conditions, SREBP has a hairpin orientation in which both the amino-terminal basic helix loop helix transcription factor motif and the carboxy-terminal regulatory domain are faced to the cytoplasm, while SCAP and INSIG bind to each other and form a hydrophobic cavity for sterol. (**B**) In conditions in which the sterol concentrations are low, the binding of SCAP and INSIG gets lost, and SCAP undergoes a conformational change that exposes a MELADL motif which is recognized by COPII. This protein serves as an escort protein, moving SCAP and bound SREBPs to the surface of the Golgi apparatus. (**C**) There, SREBP is proteolytically cleaved by the site-1 protease in the 30 amino acid stretch that links the transmembrane domains. (**D**) The site-2 protease cleaves SREBP within the membrane-spanning helix. This leads to the release of the carboxyl-terminal part of SREBP which contains the bHLH-Zip motif that shuttles to the nucleus where it activates the transcription of genes that are necessary for cholesterol and fatty acid biosynthesis.

**Figure 5 ijms-25-01109-f005:**
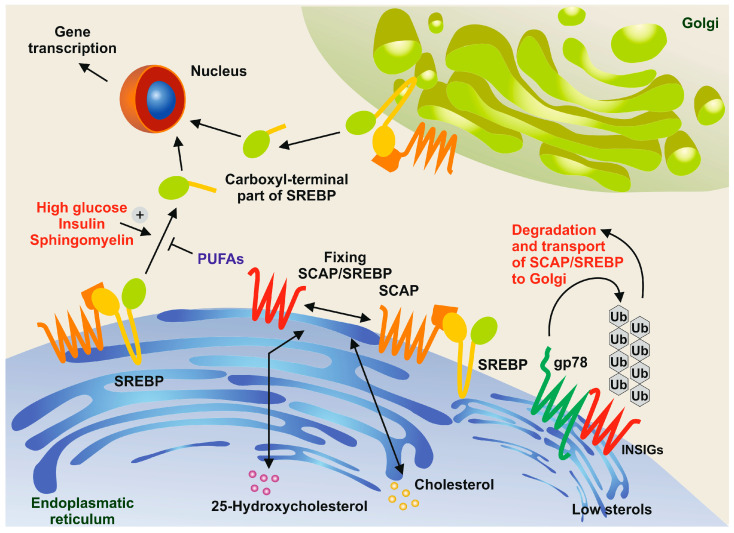
Sphingomyelin and polyunsaturated fatty acids (PUFAs) in the control of SREBP processing. In conditions with high concentrations of cholesterol and 25-hydrocholesterol, the sterol regulatory element-binding protein (SREBP) and the SREBP cleavage-activating protein (SCAP) form a complex and are retained in the endoplasmic reticulum along with insulin-induced gene proteins (INSIGs). However, when sterol concentrations are low, membrane-bound ubiquitin ligases, such as gp78, bind to INSIGs, leading to their rapid degradation through the ubiquitin pathway [48]. In turn, SREBPs are transported to the Golgi apparatus. The N-terminal domains of SREBPs are then released through the action of site-1 protease (S1P) and site-2 protease (S2P). These domains move to the nucleus, where they activate the expression of genes necessary for cholesterol and fatty acid synthesis. This proteolytic processing is triggered by high concentrations of glucose, insulin, and sphingomyelin, while it is inhibited by PUFAs.

**Figure 6 ijms-25-01109-f006:**
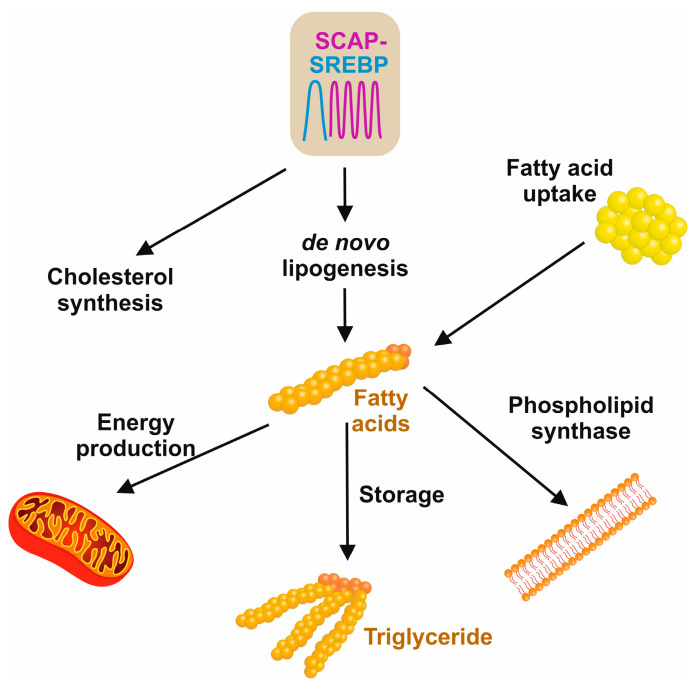
Sources and cellular usage of fatty acids. De novo lipogenesis and cholesterol synthesis are driven by the SCAP/SREBP axis. The pool of fatty acids is further increased by dietary free fatty acids that are released by the hydrolysis of triglycerides. In the cell, fatty acid is used for the synthesis of phospholipids in membrane biogenesis, mitochondrial fatty acid β-oxidation to maintain cellular energy metabolisms, or the synthesis of triglycerides that, in excess of fatty acids, can be stored in the liver or in fat cells to supply the body with energy when it is required.

**Figure 7 ijms-25-01109-f007:**
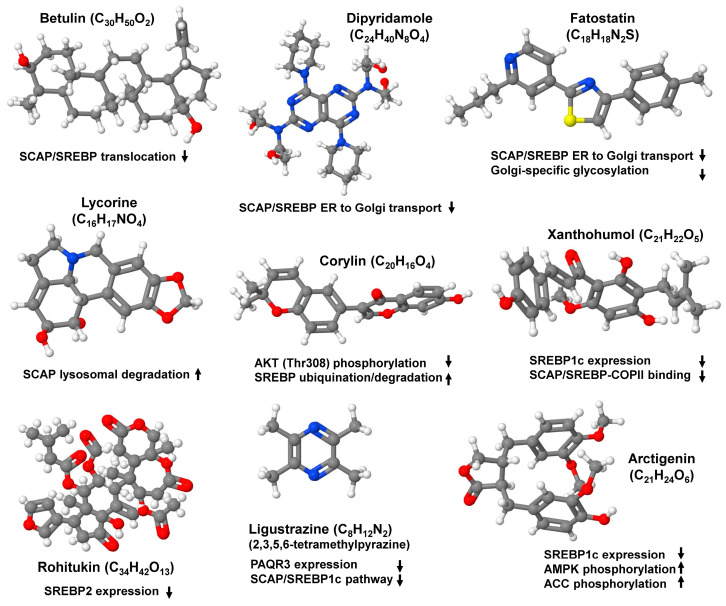
Small molecule inhibitors targeting the SCAP/SREBP axis. There are several small molecule inhibitors available that target the SCAP/SREBP axis and can be considered for the treatment of non-alcoholic fatty liver diseases. They either inhibit the translocation of the SCAP/SREBP complex from the endoplasmic reticulum to the Golgi, prevent Golgi-specific glycosylation, induce lysosomal degradation of SCAP, prevent phosphorylation of critical mediators driving lipogenic processes, inhibit SREBP expression, prevent binding of COPII to the preformed SCAP/SREBP complex, or target individual compounds of the pathways responsible for hepatic fat accumulation. Abbreviations used are: ACC, acetyl-CoA carboxylase; AMPK, AMP-activated protein kinase; PAQR3, progestin and adipoQ receptor 3; SCAP, SREBP cleavage activating protein; SREBP, sterol element binding protein.

## Data Availability

This review only presents data that were previously published. No new data were generated.

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
