# Peer review of "The Role of SCAP/SREBP as Central Regulators of Lipid Metabolism in Hepatic Steatosis"

_ijms, 2024, doi:10.3390/ijms25021109_

Round 1

Reviewer 1 Report

Comments and Suggestions for Authors

The manuscript entitled "The Role of SCAP/SREBP as Central Regulators of Lipid Metabolism in Hepatic Steatosis" provides a comprehensive exploration of the versatile roles of SCAP/SREBP regulation in de novo lipogenesis and the structural and molecular features of SCAP/SREBP in the progression of hepatic steatosis. Overall, the manuscript is well written and organized. However, there are several issues that need to be addressed for further improvement.

1.     The term "NAFLD" should be replaced with "MASLD: metabolic dysfunction-associated steatotic liver disease." Authors are requested to update this information throughout the manuscript.

2.     In the section "2. Structure of SCAP," where the authors mention the SCAP/INSIG/25-hydroxyl cholesterol ternary complex, more information about 25HC should be added. This addition will contribute to a more in-depth understanding of the complex.

3.     In the section "4. Regulation of de novo Lipogenesis by SCAP/SREBP1c," it is recommended to include a new figure that describes the regulation mechanism of SREBPs on lipid metabolism. This addition will enhance the readability and make the content more accessible to readers.

4.     The references need to be updated to resolve duplicate order numbers. Authors should carefully review and correct the reference section to ensure accuracy and avoid confusion.

Author Response

Dear Reviewer 1,

Thank you for taking the time to read our manuscript. We appreciate your helpful comments. Please find our response to your comments and suggestions in the attached pdf-file.

Regards

Ralf Weiskirchen

Reviewer 2 Report

Comments and Suggestions for Authors

The manuscript provides a comprehensive overview of the significance of the SREBP signal in the development of hepatic steatosis and MASLD, previously known as NAFLD. The manuscript is well-composed, offering valuable insights. However, a few minor errors were identified, and there are opportunities for further discussion to enhance the manuscript.

Minor Corrections:

1. It is recommended to update the terminology from NAFLD and NASH to their current nomenclature, MASLD and MASH

2. Spelling Correction Line 59: “NSIG1” -> “INSIG1”

Further Discussion:

1. How much DNL is contributing to hepatic steatosis compared to dietary lipid and adipose lipolysis during MASLD progression.

2. How SREBP coordinates with other lipid-regulating receptors, such as PPAR and LXR.

3. There are several ongoing clinical trials targeting de novo lipogenesis to reduce steatosis. How targeting SCAP specifically, as part of the SREBP pathway, could potentially offer greater effectiveness?

4. Are there any potential roles of SREBP in other liver cell types, such as inflammatory cells and hepatic stellate cells, that contribute to MASLD progression.

Author Response

Dear Reviewer 2,

Thank you for taking the time to read our manuscript. We appreciate your helpful comments. Please find our response to your comments and suggestions in the attached pdf-file.

Regards

Ralf Weiskirchen

Round 2

Reviewer 1 Report

Comments and Suggestions for Authors

The authors have addressed my previous concerns.